# Facile Preparation of Gold-Decorated Fe_3_O_4_ Nanoparticles for CT and MR Dual-Modal Imaging

**DOI:** 10.3390/ijms19124049

**Published:** 2018-12-14

**Authors:** Jing Cai, Yu Qing Miao, Li Li, Hai Ming Fan

**Affiliations:** 1Key Laboratory of Synthetic and Natural Functional Molecule Chemistry of the Ministry of Education, College of Chemistry and Materials Science, Northwest University, Xi’an 710069, China; caijing@sysucc.org.cn (J.C.); miaoyuqing@stumail.nwu.edu.cn (Y.Q.M.); 2State Key Laboratory of Oncology in South China, Sun Yat-Sen University Cancer Center, Collaborative Innovation Center for Cancer Medicine, Guangzhou 510060, China

**Keywords:** Au-Fe_3_O_4_ nanocomposites, multifunctional nanoprobe, dual-modal MR/CT imaging

## Abstract

The development of a multifunctional nanoprobe capable of non-invasive multimodal imaging is crucial for precise tumour diagnosis. Herein, we report a facile polymer-assisted method to produce Au-Fe_3_O_4_ nanocomposites (NCPs) for the dual-modal magnetic resonance (MR) and X-ray computed tomography (CT) imaging of tumours. In this approach, amino-functionalized Au nanospheres were first obtained by surface modification of the bifunctional polymer SH-PEG-NH_2_. Hydrophilic and carboxyl-functionalized Fe_3_O_4_ nanoparticles were produced by phase transfer of reverse micelle oxidation in our previous work. The Au nanoparticles were conjugated with hydrophilic Fe_3_O_4_ nanoparticles through an amide reaction. The obtained Au-Fe_3_O_4_ nanocomposites display a high r_2_ relativity (157.92 mM^−1^ s^−1^) and a Hounsfield units (HU) value (270 HU) at Au concentration of 8 mg/mL and could be applied as nanoprobes for the dual-modal MR/CT imaging of a xenografted tumour model. Our work provides a facile method to prepare Au-Fe_3_O_4_ nanocomposites for dual-modal MR/CT imaging, and this method can be extended to prepare other multifunctional nanoparticles for multimodal bioimaging.

## 1. Introduction

It is essential to develop a suitable in vivo, non-invasive imaging approach for precision treatment and prevention in cancer medicine owing to the limitations and potential for serious complications of tissue biopsy in the traditional detection of tumour disease [1,2]. With the development of clinical imaging technology, multimodal imaging has become an important research area in recent decades because of its ability to provide more sufficient and accurate imaging information than individual modalities [3], such as Positron Emission Tomography (PET)/ Computed Tomography (CT) [4], PET/ Magnetic resonance imaging (MRI) [5], CT/MRI [6], and PET/ Single Photon Emission Computed Tomography (SPECT) [7]. Among them, dual-modal CT/MR imaging has attracted particular interest in biomedical research because of the decreased radiation exposure [8]. Computed tomography (CT) can give high-resolution 3D structural details of tissues, but its low sensitivity and the small density differences in soft tissues limits the use of CT to detect tumour localization and evaluate progress [9]. In contrast, magnetic resonance (MR) imaging has lower resolution but superior soft-tissue contrast and uses non-ionizing radiation, allowing it to compensate for the shortfalls of CT imaging [10]. Thus, the combination of these two imaging modalities and the integration of their functions might improve the quality of tissue imaging. To improve contrast, multifunctional nanoprobes usually play an extremely important role in multimodal imaging. Therefore, it is essential to explore bifunctional nanoprobes with good performance in CT/MR imaging applications.

Based on the rapid development of nanotechnology, many multicomponent nanosystems have already been used as dual-modal nanoprobes for CT and MRI, such as Au-Gd hybrid [11,12], Au-Fe_3_O_4_ hybrid [13,14] and upconversion nanoparticles [15]. However, gadolinium and upconversion nanoparticles can cause acute kidney injury, chronic kidney disease [16], pneumonitis and acute inflammation [17], resulting in the potential for long-term toxicity [18]. Hence, Au and Fe_3_O_4_ nanoparticles have generally been considered the most important compositions for dual-modal agents because of their good physical performance and biocompatibility [19,20,21]. Au and Fe_3_O_4_ can be combined by two types of methods: The first is the direct synthesis of Au-Fe_3_O_4_ heterostructure nanoparticles [13,22]. However, the morphology and physical properties of such nanoparticles generally cannot be well controlled in this type of synthesis process due to lattice mismatch between Au and Fe_3_O_4_ during growth [14]. To obtain Au-Fe_3_O_4_ composites with good morphology and physical properties, it is better to synthesize the two materials in their own systems and conditions [23]. The second method is the conjugation of the two types of as-prepared nanoparticles by molecular interactions. However, there are still issues associated with Au-Fe_3_O_4_ synthesis regarding particle uniformity in terms of size and morphology, because the traditional synthesis procedure is a complicated multi-step process and cannot be controlled well [24]. Thus, developing a convenient and cost-effective procedure for the preparation of Au-Fe_3_O_4_ nanocomposites is quite desirable.

In this study, we employed the polymer SH-PEG-NH_2_ with bifunctional groups to conjugate as-transferred Fe_3_O_4_ nanoparticles and Au nanoparticles in the aqueous phase. The as-transferred Fe_3_O_4_ nanoparticles prepared by reverse micelle oxidation in our previous report have terminal carboxyl groups for further functionalization [25]. Because of the excellent chemical affinity of Au and S, Au nanoparticles could also be modified with amino groups on their surface [26]. Then, Au-Fe_3_O_4_ nanocomposites could be formed by the acetylation of terminal amines with carboxyl groups. The characteristics of the as-prepared Au-Fe_3_O_4_ nanocomposites were measured to confirm the structure, dispersibility, size and other properties by TEM, DLS, UV-vis spectroscopy, etc. The cytocompatibility was then evaluated by cell viability analysis. The potential to use Au-Fe_3_O_4_ nanocomposites as bifunctional probes for dual-modal CT/MR tumour imaging has also been explored.

## 2. Results and Discussion

### 2.1. Synthesis and Characterization of Au-Fe_3_O_4_ NCPs

Au nanospheres 60 nm in diameter were synthesized in solution through the reduction of HAuCl_4_ by NaBH_4_. To obtain the amino-functionalized gold nanospheres, the bifunctional polymer SH-PEG-NH_2_ was used to cover the surface of nanoparticles by ligand exchange and to form Au-S covalent binding. Fe_3_O_4_ nanoparticles 10 nm in diameter were synthesized by a thermal decomposition strategy in the organic phase. To achieve the carboxyl functionalization of Fe_3_O_4_ nanoparticles and enable their dispersion in aqueous solution for further binding, a phase-transfer strategy via a reverse-micelle-based oxidative reaction was performed as described in our previous work. Then, the carboxyl-functionalized Fe_3_O_4_ nanoparticles were activated and conjugated with amino-functionalized Au nanospheres by a condensation reaction. This strategy is shown schematically in Scheme 1. After vigorous washing, the product was redispersed in aqueous solution. The original colours of Fe_3_O_4_ and Au nanoparticles are dark brown and reddish purple, respectively, while the colour of Au-Fe_3_O_4_ nanocomposites changed to reddish brown after the conjugation process, indicating that the nanocomposites inherited their parental colorimetric characteristics.

The morphology of the Au-Fe_3_O_4_ nanocomposites was characterized by transmission electron microscopy (TEM), selected area electron diffraction (SAED) and energy-dispersive analysis of X-rays (EDAX), as shown in Figure 1. We can see that the 60 nm Au nanoparticles are well surrounded by the 10 nm Fe_3_O_4_ nanoparticles, and the ratio of Fe_3_O_4_ to Au is approximately 15:1. The high-resolution TEM images show that the Au and Fe_3_O_4_ nanoparticles are in close proximity. In addition, their lattice spacing is consistent with the spacing of the (311) lattice planes of the Fe_3_O_4_ particles and the (200) lattice planes of the Au particles. The EDAX of the nanocomposites further verifies the elemental composition, as Fe and Au can easily be observed in the graph. The presence of Cu, C and O is attributed to the copper grid and carbon film. In addition, this method can also be used to combine Au and Fe_3_O_4_ nanoparticles in other sizes. As it can be seen from Appendix A, the 30 nm Au nanoparticles are also well combined with the 10 nm Fe_3_O_4_ nanoparticles, and the ratio of Fe_3_O_4_ to Au is approximately 2:1. Hence, our method has universality, which can be applied to prepare a variety of Au-Fe_3_O_4_ nanocomposites by using different components.

The characteristics of Au-Fe_3_O_4_ nanocomposites are shown in Figure 2. The UV-vis spectra in Figure 2a represent the different nanoparticles. The Au nanoparticles show a main plasmon band at 520 nm, and Fe_3_O_4_ nanoparticles show a wide band at approximately 300–400 nm. After conjugation, the nanocomposites show a weak and broad plasmon band at 520 nm and a broad absorption at 300–400 nm, reflecting a change in the local electric field due to the presence of Fe_3_O_4_ nanoparticles. Figure 2b shows the hydrodynamic size and zeta potential of different types of nanoparticles. The zeta potential of the original Au nanoparticles is approximately −40 eV because of the presence of negatively charged molecules such as citric acid on the surface during synthesis in aqueous solution. After amino group functionalization, the zeta potential became +18 eV. With the introduction of carboxyl-functionalized Fe_3_O_4_ nanoparticles (zeta potential is −35 eV), the surface charges of the final product nanocomposites became negative again, indicating that the Fe_3_O_4_ nanoparticles were successfully combined with Au nanoparticles. Figure 2c shows that the hydrodynamic sizes of Au nanoparticles, amino-functionalized Au nanoparticles and Au-Fe_3_O_4_ nanocomposites are 60.28 nm, 96.80 nm and 123.63 nm, respectively. The gradual increase in these sizes indirectly reflected the successful conjugation of Au and Fe_3_O_4_ nanoparticles. The size stability of Au-Fe_3_O_4_ nanocomposites was also estimated by DLS on different days after storage. From the graph in Figure 2d, we can see that the hydrodynamic size of nanocomposites after storage on different days remained almost the same. All the results above showed that the nanocomposites prepared by our strategy were stable, well dispersible in water and suitable in size for further bioapplication.

### 2.2. T_2_ MR Relaxivity and X-ray Attenuation Property

To explore the potential of the Au-Fe_3_O_4_ nanocomposites for use in dual-modal MR/CT imaging, the T_2_ relaxivity and X-ray attenuation properties of the nanocomposites were measured. The T_2_ relaxivity of Au-Fe_3_O_4_ nanocomposites with different Fe concentrations was measured and is shown in Figure 3a. The result shows that as the Fe concentration in samples increases, the T_2_ MR signal intensity decreases, and the spots become darker. Because the Au nanoparticles were conjugated with the Fe_3_O_4_ nanoparticles via the functional polymer, the characteristics of the two types of nanoparticles did not affect each other. The T_2_ relaxivity of Au-Fe_3_O_4_ nanocomposites is approximately 157.92 mM^−1^ s^−1^, illustrating that it serve as a good T_2_ contrast agent.

The CT imaging capacity was estimated by the X-ray attenuation property of the nanocomposites. Figure 3b shows that the X-ray absorbance of the nanoparticles increased strongly as the Au concentration increased in a well linear correlation. The Hounsfield unit (HU) value revealed a well linear correlation between the Au concentration and CT attenuation. The Hounsfield units (HU) value is 270 HU at Au concentration of 8 mg/mL. It can be concluded that the Au-Fe_3_O_4_ nanocomposites at Au concentration of 135.6 mg/mL have an equivalent 4500 Hounsfield units (HU) value with eXIA™160 (corresponding to 160 mg I/mL) [27]. Hence, the Au-Fe_3_O_4_ nanocomposites could be as a positive X-ray CT nanoprobe for in vivo imaging.

### 2.3. Cytotoxicity Assays

It is essential to measure the cytotoxicity of nanocomposites for further biomedical application. The cell viability was examined by using a 3-4,5-dimethyl-thiazol-2-yl-2,5-diphenyltetrazolium bromide (MTT) assay (as shown in Figure 4). The cells treated with Au-Fe_3_O_4_ nanocomposites exhibit no significant toxicity even at 100 μg/mL and after 24 incubation with a cell viability of above 80%, indicating their high biocompatibility. These results demonstrate that these PEGylated Au-Fe_3_O_4_ nanocomposites are promising candidates for biological imaging.

### 2.4. In Vivo MR and CT Imaging of Tumours

The Au-Fe_3_O_4_ nanocomposites were used as nanoprobes for the dual-modal MR/CT imaging of a xenografted tumour model. T_2_-weighted MR imaging of the tumour-bearing mouse was performed before and after the injection of nanoprobes. As shown in Figure 5a,c, the axial and coronal scans in T_2_ MR imaging show the anatomic structure of the mouse and the profile of the tumour. The tumour MR signal intensity becomes darker than before injection. This result suggests that our Au-Fe_3_O_4_ nanocomposites can be used as nanoprobes for the MR imaging of tumours.

CT images were also acquired before and after the injection of Au-Fe_3_O_4_ nanocomposites in tumour mouse models. Compared with the control group (before injection), the CT value of the tumours treated with Au-Fe_3_O_4_ nanocomposites increased greatly. The CT imaging results are consistent with the MR imaging data. Our results revealed that the Au-Fe_3_O_4_ nanocomposites can be an effective probe for dual-modal MR/CT imaging of tumours.

## 3. Materials and Methods

### 3.1. Materials

Chemicals for the synthesis and modification of nanoparticles were purchased from Sigma-Aldrich (St. Louis, MO, USA) or Alfa-Aesar (Ward Hill, MA, USA). All other chemicals were purchased from Sinopharm Chemical Reagent Co., Ltd. (Shanghai, China). All chemical agents were used as received without further purification.

### 3.2. Synthesis of Carboxyl-functionalized Fe_3_O_4_ Nanoparticles

Uniform, monodispersed 10 nm magnetic nanoparticles coated with oleic acid were synthesized by a previously reported method [28]. The Fe_3_O_4_ nanoparticles were transferred to water and functionalized with carboxyl groups by reverse micelle oxidation [25]. Briefly, iron−oleate complex (8 mmol) and oleic acid (4 mmol) were dissolved in 1-octadecene (40 g). The mixture was heated to 320 °C and kept for 30 min. After that, the mixture was cooled to room temperature. The nanoparticles were obtained by centrifugation separation. Then, the nanoparticles (1 mg) were dispersed in cyclohexane (0.5 mL). Tertiary butanol (350 μL), K_2_CO_3_ solution (5%, 25 μL), PVP solution (40%, 50 μL), oxidizing agent solution (200 μL, 90 μg KMnO_4_ and 4.5 mg NaIO_4_) were added in the solution and stirred for 2 h. After reaction, the nanoparticles were washed and redispersed in water, and stored at 4 °C before use.

### 3.3. Synthesis of Amine-Terminated Au Nanoparticles

Au nanoparticles were synthesized following the method reported by Turkevich [29] and Frens [30]. After removal of the excess agents by centrifugation at 8000 rpm for 10 min, gold nanoparticles were redispersed in pure water to yield a final concentration of 0.01 mg/mL and stored at 4 °C before use. To obtain functionalized gold nanoparticles, 0.5 mg of NH_2_-PEG-SH (MW 2000) was added gradually to 5 mL of gold nanoparticles. The colloidal solution was mixed and stirred for 3 h at room temperature and then centrifuged at 8000 rpm for 10 min. The amine group-terminated gold nanoparticles were redispersed in water for the next process.

### 3.4. Formation of Au-Fe_3_O_4_ Nanocomposites

In the formation process, 1 mL of carboxyl-modified Fe_3_O_4_ nanoparticles (1 mg/mL) was centrifuged and re-suspended in 10 mM MES buffer (pH 5.5). Then, 100 µL of EDC (4 mg/mL) and 100 µL of NHS (6 mg/mL) were added to the Au nanoparticle solution and sonicated at 4 °C for 30 min. Then, 2 mL of PEGylated amine-modified gold nanoparticles was added to the activated Fe_3_O_4_ nanoparticle solution and stirred for 2 h. The resulting solution was centrifuged at 8000 rpm for 10 min to remove unbound magnetic nanoparticles, and then the free gold nanoparticles without attached Fe_3_O_4_ were separated and removed under an external magnetic field.

### 3.5. Characterization

The TEM images were taken by using a JEM-2010HR transmission electron microscope (JEOL, Tokyo, Japan) with a tungsten filament at an accelerating voltage of 200 kV. High-resolution transmission electron microscopy (HR-TEM) and energy-dispersive X-ray analysis spectroscopy (EDAX) were performed on an FEI Tecnai G2 F30 transmission electron microscope (at 300 kV, FEI, USA). Magnetic measurements were carried out on a magnetic property measurement system (MPMS XL-7, Quantum Design, San Diego, USA). The UV-vis absorption of different nanoparticle samples was measured with a UV-vis-NIR spectrophotometer (UV-3150, Shimadzu, Japan). The hydrodynamic size and surface potential of the nanoparticles were determined in aqueous phase by using a Malvern Zetasizer Nano-ZS (Malvern Instruments, Worcestershire, UK). The r_2_ relaxivity was determined by a linear fitting of 1/T_2_ as a function of the Fe concentration of the particles. The instrumental parameters were set as follows: point resolution of 156 × 156 mm^2^, section thickness of 2 mm, TR of 5000 ms and number of signal acquisitions of 3.

### 3.6. Cytotoxicity Assay

Nasopharyngeal epithelium carcinoma CNE2 cells were cultured in RPMI 1640 medium containing heat-inactivated FBS (10%, *v*/*v*). An MTT assay was used to evaluate the viability of the cells treated with the Au-Fe_3_O_4_ nanocomposites. First, CNE2 cells were seeded in a 96-well plate at a density of 1 × 10^4^ cells per well. After overnight incubation and adherence, the medium was replaced with fresh medium containing Au-Fe_3_O_4_ nanocomposites at different concentrations and at various incubation times. After incubation, MTT in PBS was added to each well to a final concentration of 50 μg/mL) for 3 h. The optical density was measured at 570 nm in a microplate reader. The cell survival was expressed as the percentage of absorption of the treated cells compared with that of the control cells (no nanocomposites present during incubation). One-way ANOVA statistical analysis with post hoc testing was used to evaluate the significance of the data. Probability levels less than 0.05 were taken to demonstrate significant differences, and the data were indicated by (*) for *p* < 0.05, (**) for *p* < 0.01, and (***) for *p* < 0.001, respectively.

### 3.7. In Vivo CT/MR Imaging of a Xenografted Tumour Model

In vivo experiments were carried out according to protocols approved by the institutional committee for animal care (Approval No. SYSU-IACUC-2018-000028). The CNE2 tumour xenograft model was established in male 4–6-week-old Balb/c nude mice by subcutaneously injecting 2 × 10^7^ CNE2 cells into the right flank region. When the tumour nodules reached a volume of 0.5–1 cm^3^, the mice were anaesthetized and allocated to the control and Au-Fe_3_O_4_ nanocomposite groups. Au-Fe_3_O_4_ nanocomposites were injected into the tumours for dual imaging. CT and MR scans were performed both before and after injection by a GE Discovery CT750 HD clinical imaging system (120 kV) and an MR clinical system (Siemens Trio, Erlanen, Germany) with a custom-built rodent receiver coil. The 2D spin-echo T_2_-weighted MR images were obtained with 2 mm slice thickness, 4200/80 ms TR/TE, 192 × 320 mm^2^ FOV, and NEX = 8.

## 4. Conclusions

A unique approach to preparing Au-Fe_3_O_4_ nanocomposites for the dual-modal CT/MR imaging of tumours is developed. As-prepared Au nanospheres conjugate with carboxyl-functionalized Fe_3_O_4_ nanoparticles transferred from water, via chemical bond linkage with the bifunctional polymer SH-PEG-NH_2_. The prepared Au-Fe_3_O_4_ nanocomposites show enhanced X-ray attenuation and non-compromised r_2_ relaxivity (157.92 mM^−1^ s^−1^). According to the cell viability measurement, as-prepared Au-Fe_3_O_4_ nanocomposites also show good cytocompatibility in the given concentration range. Importantly, the Au-Fe_3_O_4_ nanocomposites have excellence T_2_ and CT performance and can be used as efficient nanoprobes for the dual-modal CT/MR imaging of xenografted tumour models. The multifunctional Au-Fe_3_O_4_ nanocomposites may have great potential for use as nanoprobes in the dual-modal CT/MR imaging of tumours.

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
