# Peer review of "Facile Preparation of Gold-Decorated Fe3O4 Nanoparticles for CT and MR Dual-Modal Imaging"

_ijms, 2018, doi:10.3390/ijms19124049_

Round 1

Reviewer 1 Report

The paper is more interesting for the development of new synthesis method in order to obtained Au-Fe3O4 nanocomposites for imaging applications. Therefore, it is recommended that this contribution be accepted after minor revision.

The following minor revision is suggested before acceptance for publication:

- pag 1, line 10 – Tel. Number is not completed

-The authors should write more specifically the experimental part of nanoparticles preparation. It’s very poor.
- In the Results and discussion section page 3 line 89 “…Transition” please changed with Trasmission.

- The authors used MTT assay to the test cell viability of NPs,  it’s necessary measure the cell viability to different time points.

Author Response

Thanks for your comments.

 1. pag 1, line 10 – Tel. Number is not completed

- We added the Tel. number instead of -xxxx on pag 1, line 10.

2. The authors should write more specifically the experimental part of nanoparticles preparation. It’s very poor.

- We added a reference and showed the experimental details in nanoparticles preparation. We also corrected some mistakes in this part. Added reference: Ref 26 [Nat. Mater. 2004, 3, (12), 891-5].

3. In the Results and discussion section page 3 line 89 “…Transition” please changed with Trasmission.

- We change the word to “trasmission” in page 3 line 90.

4. The authors used MTT assay to the test cell viability of NPs, it’s necessary measure the cell viability to different time points.

- We perform the experiments of MTT assay on different time points. The results are shown in Figure 4.

Reviewer 2 Report

The area of Au/Fe, Fe2O3 or Fe3O4 nanocomposites is a well-known field, so I do not see a serious novelty in this work, except the modification of synthesis method. Such nanoparticles can be core-shell (Au core - iron oxide shell) or Au core with iron oxide nanoparticles on its surface. 

The manuscript is technically good.

The reference list needs to include main books and reviews in the field of iron- and gold-containing nanomaterials.

Author Response

Thanks for your comments.

The reference list needs to include main books and reviews in the field of iron- and gold-containing nanomaterials.

- We added two references in the modified manuscript and we also changed the order in References. Ref 20 [Chem. Soc. Rev. 2012, 41, (5), 1911-28]. Ref 21 [Chem. Soc. Rev. 2015, 44, (23), 8410-23].

Reviewer 3 Report

Line 57: It might be better to change the sentence to: “However, there are still issues associated with Au-Fe3O4 synthesis regarding particles uniformity in size and morphology, …..”

Line 98: “As we can see”: it might be better to use passive tense

The brand and model of DLS used in this study are missing.

As Fe3O4 NPs are magnetic, one of the main issues in formulation and administration is aggregation and low stability of the suspended particles. DLS normally shows a very unrealistic and high hydrodynamic size in such particles, hence TEM is required to identify the precise particle size. Figure 2 (c) shows similar particle size as TEM result. More explanation is required to clarify how the aggregation was prevented and high particle stability in aqueous system was achieved over 20 days (Figure 2 (d)).

Line 135: It might be better to change “It is clear that” to “The result shows that”

Figure 3 Legend is not in the same page as the figure.

Line 150: The last sentence does not have a verb.

Line 172: The particles were directly injected into the tumor and therefore there is no tumour targeting applied but the paper mentioned “Our results revealed that the Au-Fe3O4 nanocomposites can target tumours after injection, allowing the effective dual-modal MR/CT imaging of tumours”. There’s no proof for that statement.

Line 194 and 198: mL and milliliter unit mentioned respectively (Consistency in units required).

Line 218: CNE2 cells need to be defined.

Author Response

Thanks for your comments.

1. Line 57: It might be better to change the sentence to: “However, there are still issues associated with Au-Fe3O4 synthesis regarding particles uniformity in size and morphology, …..”

-We change the sentence in line 57.

2. Line 98: “As we can see”: it might be better to use passive tense

-We change to “As it can be seen” in line 100.

3. The brand and model of DLS used in this study are missing.

-We add the model of DLS instrument in line 223.

4. Line 135: It might be better to change “It is clear that” to “The result shows that”

-We change the words in line 135.

5. Figure 3 Legend is not in the same page as the figure.

-We change the typesetting.

6. Line 150: The last sentence does not have a verb.

-We add the word “be” in line 153.

7. Line 172: The particles were directly injected into the tumor and therefore there is no tumour targeting applied but the paper mentioned “Our results revealed that the Au-Fe3O4 nanocomposites can target tumours after injection, allowing the effective dual-modal MR/CT imaging of tumours”. There’s no proof for that statement.

-We change the sentence to “Our results revealed that the Au-Fe3O4 nanocomposites can be an effective probe for dual-modal MR/CT imaging of tumours.” in line 174.

8. Line 194 and 198: mL and milliliter unit mentioned respectively (Consistency in units required).

-We change the word “milliliter” to “mL” in 201

9. Line 218: CNE2 cells need to be defined.

-We define the CNE2 cells on its first occurrence.